# Effect of Serum Lipid Profile on the Risk of Breast Cancer: Systematic Review and Meta-Analysis of 1,628,871 Women

**DOI:** 10.3390/jcm11154503

**Published:** 2022-08-02

**Authors:** Mehran Nouri, Mohammad Ali Mohsenpour, Niki Katsiki, Saeed Ghobadi, Alireza Jafari, Shiva Faghih, Maciej Banach, Mohsen Mazidi

**Affiliations:** 1Department of Community Nutrition, School of Nutrition and Food Sciences, Shiraz University of Medical Sciences, Shiraz 7193635899, Iran; mehran_nouri71@yahoo.com (M.N.); shivafaghih@gmail.com (S.F.); 2Students’ Research Committee, School of Nutrition and Food Science, Shiraz University of Medical Sciences, Shiraz 7193635899, Iran; mohammadali.mohsenpour@gmail.com; 3Department of Clinical Nutrition, School of Nutrition and Food Sciences, Shiraz University of Medical Sciences, Shiraz 7134845794, Iran; 4Department of Nutritional Sciences and Dietetics, International Hellenic University, 574 00 Thessaloniki, Greece; nikikatsiki@hotmail.com; 5School of Medicine, European University Cyprus, Nicosia 2404, Cyprus; 6Non-Communicable Diseases Research Center, Alborz University of Medical Sciences, Karaj 3149969415, Iran; sghobadi70@gmail.com; 7Department of Community Nutrition, School of Nutritional Sciences and Dietetics, Tehran University of Medical Sciences, Tehran 1717613151, Iran; jafari.alireza1975@gmail.com; 8Department of Hypertension, Chair of Nephrology and Hypertension, WAM University Hospital in Lodz, Medical University of Lodz, Zeromskiego 113, 90-549 Lodz, Poland; 9Polish Mother’s Memorial Hospital Research Institute (PMMHRI), 93-338 Lodz, Poland; 10Cardiovascular Research Centre, University of Zielona Gora, 65-417 Zielona Gora, Poland; 11Clinical Trial Service Unit and Epidemiological Studies Unit (CTSU), Nuffield Department of Population Health, University of Oxford, Oxford OX1 2JD, UK; 12Department of Twin Research & Genetic Epidemiology, King’s College London, South Wing St Thomas’, London SE1 7EH, UK

**Keywords:** breast cancer, total cholesterol, triglycerides, low-density lipoprotein cholesterol, high-density lipoprotein cholesterol, apolipoprotein A

## Abstract

Dyslipidemia has been linked to breast cancer incidence. The aim of the present meta-analysis was to further investigate the relationship between the serum lipid profile and breast cancer risk. Databases such as PubMed, EMBASE, and Web of Sciences were searched up to the end of January 2021 using certain MeSH and non-MeSH keywords and combinations to extract related published articles. Twenty-six prospective studies involving 1,628,871 women, of whom 36,590 were diagnosed with breast cancer during the follow-up period met the inclusion criteria. A negative and significant association was found between the HDL-C level and the risk of breast cancer (relative risk (RR): 0.85, 95% CI: 0.72–0.99, I^2^: 67.6%, *p* = 0.04). In contrast, TG (RR: 1.02, 95% CI: 0.91–1.13, I^2^: 54.2%, *p* = 0.79), total cholesterol (TC) (RR: 0.98, 95% CI: 0.90–1.06, I^2^: 67.2%, *p* = 0.57), apolipoprotein A (ApoA) (RR: 0.96, 95% CI: 0.70–1.30, I^2^: 83.5%, *p* = 0.78) and LDL-C (RR: 0.93, 95% CI: 0.79–1.09, I^2^: 0%, *p* = 0.386) were not associated with breast cancer development. In studies adjusting for hormone use and physical activity, breast cancer risk was positively correlated with TC (RR: 1.05, 95% CI: 1.01–1.10). Similarly, TG was significantly related to breast cancer development after adjustment for baseline lipids (RR: 0.92, 95% CI: 0.85–0.99) and race (any races mentioned in each study) (RR: 1.80, 95% CI: 1.22–2.65). In the present meta-analysis, HDL-C was inversely related to breast cancer risk. Overall, data on the links between lipids and breast cancer are conflicting. However, there is increasing evidence that low HDL-C is related to an increased risk for this type of malignancy.

## 1. Introduction

Breast cancer is the most frequently diagnosed tumor in women according to the World Health Organization (WHO) [1]. According to the Centre for Disease Control (CDC), there are several predisposing factors to breast cancer, including older age, genetic mutations, early initiation of menstruation (at the age of <12 years), late menopause (at the age of >55 years), family history of breast or ovarian cancer, previous exposure to radiation, obesity, physical inactivity, alcohol use, and the use of hormone replacement therapy or oral contraceptives [2].

A link between dyslipidemia and breast cancer has been reported. For example, increased levels of triglycerides (TGs), low-density lipoprotein cholesterol (LDL-C) and very low-density lipoprotein cholesterol (VLDL) were observed in breast cancer patients compared with normal controls [3,4]. However, there are conflicting results [5,6].

The relationship between high-density lipoprotein cholesterol (HDL-C) and the risk for breast cancer remains unclear with some studies reporting an inverse association and others reporting the opposite or no association at all [7,8,9,10]. Furthermore, it has been proposed that HDL functionality may affect these links [7,8,9,10]. A Mendelian randomization study found that genetically elevated plasma LDL-C and HDL-C levels were associated with an increased breast cancer risk [11]. Of note, 27-hydroxycholesterol can facilitate metastasis via the induction of estrogen-receptor-positive breast cancer cells [12]. Furthermore, HDL glycation and oxidative modification of lipoproteins may activate certain inflammation-related pathways, leading to cell proliferation and migration, as well as inhibiting apoptosis [12]. Overall, the lipid profile may be a predictor of breast cancer occurrence and recurrence [13].

The aim of the present systematic review and meta-analysis was to further investigate the relationship between the serum lipid profile and breast cancer development.

## 2. Materials and Methods

The Preferred Reporting Items for Systematic Reviews and Meta-Analyses (PRISMA) guidelines were followed to conduct the present systematic review and meta-analysis. This meta-analysis is registered in the International Prospective Register of Systematic Reviews (PROSPERO) under registration number CRD42021281278.

## 3. Search Strategy

We searched for papers published up to the end of January 2021 in databases such as PubMed, EMBASE, and Web of Sciences (ISI). The search strategy used MeSH and non-MeSH keywords and combinations to extract related published articles. The keywords used were “TC”, “HDL-C”, “LDL-C”, “TG”, “Apolipoprotein”, “lipoproteins”, “cholesterol”, “triglyceride”, “dyslipidemias”, “lipid profile”, “lipid component”, “blood lipid”, “plasma lipid”, “serum lipid”, “plasma lipoprotein, “dyslipoproteinemia”, “hypercholesterolemia”, “hypertriglyceridemia”, “hyperlipidemia”, “lipemia”, “ApoA”, “Apolipoproteins A”, “ApoB”, “Apolipoproteins B”, and “Metabolic syndrome”. Breast cancer was defined using the terms “breast neoplasm”, “breast cancer”, “breast tumor”, “breast tumour”, “breast malignancy”, and “breast carcinoma”. Furthermore, “Cohort”, “Prospective”, “Longitudinal”, “Follow-up”, “Nested”, and “Population-based” terms were used to limit the findings to cohort studies. No automatic filtering of databases was used during the database search. The references of papers were also checked. No time or language limitations were applied. The reference list of eligible articles was further searched, and authors were contacted by email for additional data, if needed.

## 4. Eligibility Criteria

Original articles that fulfilled the following criteria were included in the present systematic review and meta-analysis: (1) cohorts with a prospective design (the exposure takes place before the outcome), (2) participants free of cancer at baseline, and, (3) investigations of the relationship between the lipid profile and the risk of developing breast cancer.

After excluding duplicates and based on titles and abstracts, we excluded animal studies and those involving humans aged ≤ 18 years. In addition, supplementary hand searching of the reference lists of previous reviews or meta-analyses was conducted. Of 93 eligible full articles, 26 articles met the inclusion criteria (Figure 1).

## 5. Study Selection

Study selection started with the removal of duplicates, followed by the screening of titles and abstracts by two reviewers (M.N and M.A.M) blinded to the names, qualifications and institutional affiliations of the study authors. Agreement between the reviewers was excellent (Kappa index: 0.86; *p* < 0.001). Disagreements were resolved at a meeting between reviewers prior to selected articles being retrieved (a flow chart is available in Figure 1). We included studies if they met all of the following criteria: (1) the outcome of interest was the lipid profile; (2) the studies were population-based cohort studies and reported breast cancer data; (3) relative risk (RR), hazard ratio (HR) or odds ratio (OR) estimates with 95% confidence interval (CI) adjusted for multivariable factors were available or could be calculated; and (4) articles were original with full-text in English. Studies were excluded according to the following criteria: (1) reviews, letters, opinion papers, editorials, unpublished data or comments; (2) those published in languages other than English; (3) those that were not population-based cohort studies; or (4) RR, HR or OR estimates with 95% CI were not available or could not be calculated. Publications lacking primary data and/or explicit method descriptions were also excluded.

## 6. Data Extraction and Management

The full text versions of studies meeting the inclusion criteria were retrieved and screened to determine their eligibility by two reviewers (S.G and A.R.J). A study quality assessment was performed according to the Newcastle-Ottawa Scale (NOS, Table 1 [8,9,14,15,16,17,18,19,20,21,22,23,24,25,26,27,28,29,30,31,32,33,34,35,36,37]) as bias-assessment scores without an effect on selection [38]. By evaluating the selection, comparability and outcome of each study, the rating system scored studies from 0 (highest degree of bias) to 9 (lowest degree of bias). Furthermore, we investigated the funding sources of all eligible studies. Following an assessment of the methodological quality, two reviewers (S.G. and A.R.J.) extracted data using a purpose-designed data extraction form and wrote independent summaries on what they considered to be the most important results from each study. These summaries were compared, and any differences in opinion were resolved by discussion and consultation with a third reviewer (M.M). Any further calculations of the study data considered necessary were conducted by the first reviewer (S.G) and checked by the second (A.R.J). Information extracted from each eligible study included the following items: author, year and references, country, study name, men (%), mean age, follow-up time (years), number of cases, number of participants, parameters, outcomes and main confounders.

## 7. Data Synthesis and Statistical Analyses

We performed a random-effect meta-analysis as described by DerSimonian and Laird to estimate the summary effect size and 95% CI in the present highest vs. lowest meta-analysis. Comparisons between studies were ignored and no clear division was made [39]. We used the most fully adjusted hazard risk reported in the included studies. For studies with menopausal-specific effect sizes, we combined pre- and post-menopause estimates by employing a fixed-effects model and using the combined effect size for the analyses. Heterogeneity was tested using the Cochran Q and I^2^ statistics [40]. A series of subgroup analyses was also performed to identify potential sources of heterogeneity based on adjustments for the main confounders, including smoking, hormone, race, alcohol, body mass index, physical activity and lipid profile at baseline. The publication bias was evaluated through visual inspection of the funnel plot. A statistical assessment of the publication bias was conducted with Egger’s regression asymmetry [41] and Begg’s test [42]. All statistical analyses were performed using STATA version 16.0 (StataCorp, College Station, TX, USA), and two-sided *p* values of <0.05 were considered statistically significant.

## 8. Results

Of the 93 eligible full articles, 26 prospective studies met the inclusion criteria; their key characteristics are shown in Table 1. Among these studies, 21/26 were cohort studies [8,9,14,15,16,17,18,19,20,21,22,23,24,25,26,27,28,29,30,31,32] and 5/26 were nested case control studies [33,34,35,36,37]. Study sample sizes ranged from 594 to 288,057 persons aged >20 years. The mean follow up period was 12.41 years (range 7 to 26 years). Sixteen of the studies were conducted in European countries [8,9,14,15,16,18,24,25,26,28,29,30,32,34,35,36], 7 in the USA [17,19,20,27,31,33,37], 2 in Japan [22,23] and 1 in Korea [21]. Overall, 1,628,871 women were included in the present meta-analysis, of whom 36,590 were diagnosed with breast cancer during the follow-up period. The results of the NOS quality assessment are shown in the Table 1, with 18 studies scoring values ≥7 [8,9,17,18,19,20,23,24,25,26,27,28,29,30,31,32,35,36] and no studies scoring <5.

**Table 1 jcm-11-04503-t001:** Summary of included studies.

Author’s Name, Country, Year	Sample Size	Age (Years)	Study Design	No. of Incident Breast Cancer/Case	Follow Up (Mean, Year)	Data Presented	Adjusted	Results	Study Score
Lars J. Vatten et al. Norway (1990) [14]	24,329	31–51	Cohort	242	14	TG, TC	Age, BMI	There were no statistically significant associations of lipid measures with breast cancer risk.	5
Annette Pernille Hoyer et al. Denmark (1992) [15]	5207	30–80	Cohort	51	26	HDL, LDL, TG, TC	Age, Smoking, Menopause Age, Alcohol Intake, BMI, Socioeconomic Status	There was a significant association of HDL with breast cancer risk.	5
Maria Gaard et al. Norway (1994) [16]	30,666	20–54	Cohort	302	10.4	HDL, LDL, TG, TC	Age, Smoking, Menopause Age, BMI, Lipid Baseline	There were no statistically significant associations of lipid measures with breast cancer risk.	6
Kyle Steenland et al. USA (1995) [17]	14,407	25–74	Cohort	163	17	TC	Age, Smoking, Menopause Age, Alcohol Intake, BMI, Socioeconomic Status, Physical Activity, Parity	There was no statistically significant association of TC with breast cancer risk.	7
Patricia. G Moorman et al. USA (1998) [33]	400 (control), 196 (case)	41.3	Nested case-control	Case	_	HDL	Age, Menopause Age, Education, BMI, Alcohol Intake, Family History Cancer, Hormone Use, History of Hysterectomy, Parity, Smoking	There was no statistically significant association of HDL with breast cancer risk.	6
J Manjer et al. Sweden (2001) [18]	9738	49.6 ± 7.8	Cohort	269	13.1	TG, TC	Age, Smoking, Alcohol Intake, BMI, Parity	There were no statistically significant associations of lipid measures with breast cancer risk.	8
Anne-Sofie Furberg et al. Norway (2004) [9]	38,823	43	cohort	708	17.2	HDL	Age, Menopause Age, Smoking Socioeconomic Status, BMI, Parity, Lipid Baseline	There was no statistically significant association of HDL with breast cancer risk.	9
A. Heather Eliassen et al. USA (2005) [19]	71,921	66 (mean)	Cohort	2468	10	TC	Age, Menopause Age, Alcohol intake, BMI, Physical Activity, Parity, Family History Cancer, Hormone Use	There was no statistically significant association of TC with breast cancer risk.	8
Manami Inouea et al. Japan (2008) [22]	18,176	55.5 ± 8.1	Cohort	120	10	HDL, TG	Age, Smoking, Alcohol Intake, Lipid Baseline	There were no statistically significant associations of lipid measures with breast cancer risk.	6
Anna M. Kuchareska-Newton et al. USA (2008) [20]	7575	53.7 ± 5.7	Cohort	359	13	HDL	Age, Menopause Age, BMI, Race, Smoking Hormone Use	There was no statistically significant association of HDL with breast cancer risk.	8
Guy Fagherazzi et al. France (2009) [25]	69,088	40–65	Cohort	2932	12	TC	Menopause Age, Alcohol Intake, BMI, Family History Cancer, Hormone Use	There was no statistically significant association of TC with breast cancer risk.	7
Hiroyasu Iso et al. Japan (2009) [23]	21,685	54.2 (mean)	Cohort	178	12.4	TC	Age, Smoking, Alcohol Intake, BMI	There was a significant association of TC with breast cancer risk.	8
H Ulmer et al. Austria (2009) [24]	84,460	41.8 ± 15.1	Cohort	1204	10.6	TG	Smoking, BMI, Socioeconomic status, Lipid Baseline	There was no statistically significant association of TG with breast cancer risk.	9
Mina Ha et al. Korean (2009) [21]	170,374	40–64	Cohort	714	10	TC	Age, Age at Menarche, Age at First Childbirth, Nulliparity, Hormone Replacement Therapy, Duration of Breast Feeding, Smoking Habit, Alcohol Consumption	There was a positive association between cholesterol level and breast cancer risk.	6
Immacolata Capasso et al. Italy (2010) [35]	777 [control), 210 [case)	57.5	Nested case-control	Case	_	HDL	Age, Menopause Age, BMI, Alcohol Intake, Family History cancer, Hormone Use, Parity, Smoking, Socioeconomic status	There was a significant association of HDL with breast cancer risk.	7
C. Agnoli et al. Italy (2010) [34]	1089 (control), 163 (case)	58 ± 5.6	Nested case-control	Case	_	HDL, TG	Age, Smoking, Menopause Age, Education, Alcohol Intake, Family History Cancer, Hormone Use	There were significant associations of HDL and TG with breast cancer risk.	6
Wegene Borena et al. Norway, Austria, and Swede (2011) [26]	256,512	44.2	Cohort	5006	11.9	TG	Age, Smoking, BMI	There was no statistically significant association of TG with breast cancer risk.	8
Jennifer C. Melvin et al. Sweden (2012) [28]	234,494	25<	Cohort	6105	8.25	HDL, LDL, TG, TC, APO A, APO B, TC/HDL, LDL/HDL, TG/HD, APO B/APO A	Age, Socioeconomic Status, Lipid Baseline, Parity	There was a significant association of TG with breast cancer risk.	7
Jaclyn L. F. Bosco et al. USA (2012) [27]	49,172	21–69	Cohort	1228	10.5	TC	Age, Race, Education, BMI, Physical Activity	There was no statistically significant association of TC with breast cancer risk.	7
Susanne Strohmaier et al. Norway, Austria, and Sweden (2013) [29]	288,057	33–48	Cohort	5228	11.7	TC	Age, Smoking, BMI	There was a significant association of TC with breast cancer risk.	8
Signe Borgquist et al. Sweden (2016) [30]	17,035	57.9	Cohort	1024	14.3	APO A, APO B, APO B/APO A	Age, Menopause Age, Socioeconomic status, BMI, Hormone Use, Parity	There were significant associations of APO B/APO A with breast cancer risk.	7
Mathilde His et al. France (2017) [36]	1043 (control), 583 [case)	50–63	Nested case-control	Case	_	HDL, LDL, TG, TC, TC/HDL, LDL/HDL	Age, Menopause Age, Smoking, BMI, Family History Cancer, Education, Alcohol Intake, Hormone Use	There were no statistically significant associations of lipid measures with breast cancer risk.	7
Daniel T. Dibaba et al. USA (2018) [31]	94,555	50–71	Cohort	5380	14	TC	Age, Education, BMI, Physical Activity, Family History Cancer, Hormone Use, History of Hysterectomy, Parity, Smoking	There was a significant association of TC with breast cancer risk.	8
Kasper Mønsted Pedersen et al. Denmark (2020) [8]	56,790	57.5	Cohort	1641	7.4	HDL, APO A	Age, Smoking, BMI, Physical Activity, Education, Alcohol Intake, Socioeconomic status, Lipid Baseline	There was a significant association of Apo A with breast cancer risk.	7
Catherine Schairer et al. USA (2020) [37]	2470 (control), 247 (case)	60.7	Nested case-control	Case	_	HDL, TG, TG/HDL	Age, Race	There were significant associations of HDL and TG/HDL with breast cancer risk.	6
Rhonda S. Arthur et al. UK (2021) [32]	58,629	60 (56–64)	Cohort	1268	7	HDL, TG	Age, BMI, Physical activity, Family History Cancer, Alcohol intake, Hormone Use, Smoking, Socioeconomic status	There were no statistically significant associations of lipid measures with breast cancer risk.	7

Abbreviations: HDL–C: high-density lipoprotein cholesterol; LDL–C: low-density lipoprotein cholesterol; TG: triglyceride; TC: total cholesterol; APO: apolipoprotein; BMI: body mass index.

## 9. Associations of Lipid Profile with Risk of Breast Cancer

A negative and significant association was found between the HDL–C level and the risk of breast cancer (RR: 0.85, 95% CI: 0.72–0.99, n = 13 studies, Figure 2) with a moderate risk of heterogeneity (I^2^: 67.6, *p* = 0.04). In contrast, TG (RR: 1.02, 95% CI: 0.91–1.13, n = 12 studies, I^2^: 54.2%, *p* = 0.79, Figure 3), total cholesterol (TC) (RR: 0.98, 95% CI: 0.90–1.06, n = 14 studies, I^2^: 67.2%, *p* = 0.57, Figure 4), apolipoprotein A (ApoA1) (RR: 0.96, 95% CI: 0.70–1.30, n = 3 studies, I^2^: 83.5%, *p* = 0.78, Figure 5) and LDL–C (RR: 0.93, 95% CI: 0.79–1.09, n = 4 studies, I^2^: 0%, *p* = 0.38, Figure 6) were not associated with breast cancer development.

Of note, a small but significant positive correlation was found between TC and breast cancer risk in studies adjusting for hormone use (RR: 1.05, 95% CI: 1.01–1.10) and physical activity (RR: 1.05, 95% CI: 1.01–1.10). Similarly, TG was significantly related to breast cancer development after adjustment for baseline lipids (RR: 0.92, 95% CI: 0.85–0.99) and race (any race mentioned in each study) (RR: 1.80, 95% CI: 1.22–2.65) (added in Appendix A).

## 10. Publication Bias

Visual inspection of the funnel plot symmetry suggested no potential publication bias for the comparisons of HDL–C (Egger = 0.237), TG (Egger = 0.069), TC (Egger = 0.480) and LDL–C with the risk of breast cancer (Figure 7). Furthermore, Egger’s linear regression (intercept = −2.3, 95% CI: −6.91 to 6.00, two-tailed *p* = 0.542) and Begg’s rank correlation test (Kendall’s Tau with continuity correction =1.00, z = 0.342, two tailed *p* = 0.436) indicated the absence of a publication bias. After adjustment of the effect size for the potential publication bias using the ‘trim and fill’ correction, no potentially missing studies were added to the funnel plot for HDL–C, TG, TC or LDL–C.

## 11. Discussion

The present systematic review and meta-analysis evaluated the associations between lipid parameters and the risk of developing breast cancer in women. Only HDL-C was found to be significantly related to breast cancer development with an RR value of 0.85, thus highlighting the potential preventive effect of elevated HDL-C levels. Similarly, low HDL-C levels (<77 mg/dL) were correlated with an increased risk of breast cancer in the Copenhagen General Population Study [8]. Of note, the increased risks with lower HDL-C levels, i.e., HR, were 1.20 (95% CI: 1.01–1.41) and 1.36 (95% CI: 1.11–1.66) for HDL-C concentrations of 58–77 mg/dL and 39–58 mg/dL, respectively. A modest but non-significant, inverse association between HDL-C and breast cancer was reported in 2 previous meta-analyses produced in 2015; one by Touvier et al. which included 22 prospective cohort studies and 2 nested case-control studies (HR 0.90, 95% CI: 0.77–1.04, I^2^: 52%) [43], and the second by Ni et al., which included 15 prospective cohort studies with 1,189,635 participants and 23,369 breast cancer cases (RR 0.92, 95% CI: 0.73–1.16, I^2^: 65%) [44]. In the latter meta-analysis, the inverse association between HDL-C and breast cancer risk was significant among women who were postmenopausal at baseline (RR 0.77, 95% CI: 0.64–0.93), whereas the RR value was 0.84 (95% CI: 0.40–1.74) for premenopausal women [44]. Therefore, the potential involvement of HDL-C in breast cancer development seems more pronounced after menopause, highlighting the importance of implementing health policy strategies to increase HDL-C levels (and avoid their reduction) in postmenopausal women.

In the present meta-analysis, TC, TG and LDL-C were not found to be related to breast cancer risk. Similar results were reported in the abovementioned previous meta-analyses conducted by Touvier et al. [43] and Ni et al. [44]. Despite these findings, a high cholesterol intake has been positively related to an increased risk of breast cancer, and increased LDL receptor expression has been observed in breast cancer tissue to enhance LDL-C uptake from the circulation, since proliferating cancer cells require more cholesterol [12]. Furthermore, a recent Mendelian randomization study found that genetically elevated plasma LDL-C (OR 1.03, 95% CI: 1.01–1.07, *p* = 0.02) and HDL-C (OR 1.06, 95% CI: 1.03–1.10, *p* = 0.0049) correlated with an increased breast cancer risk [11].

As already mentioned, in the present meta-analysis, after adjustments for hormone use and physical activity, TC was found to be positively associated with breast cancer risk, as was TG when adjusted for baseline lipids and race. Overall, there are conflicting data regarding the links between lipids (especially TC, TG and LDL-C) and the risk for breast cancer [45]. For example, TGs have been considered a prognostic factor for breast cancer occurrence and recurrence, although not in all studies [45]. These contradictory findings may be attributed to the multifactorial nature of the disease and the presence of several confounding factors, such as age, race, menopausal status, obesity, genetic mutations, physical inactivity, and alcohol and hormone use. Large prospective and mechanistic studies (both in vitro and in vivo) are required to fully elucidate the role of lipids in breast cancer development in different populations. Furthermore, further investigations are warranted to investigate whether HDL-raising agents may protect against breast cancer development and progression. Of note, statins have been persistently reported to improve breast cancer outcomes and protect against the development and progression of this malignancy [46,47,48,49].

ApoA was also not significantly related to breast cancer risk in the present meta-analysis. Similarly, no association has previously been observed between ApoA1 and breast cancer incidence [30], whereas other studies have reported that ApoA1 is a risk factor for intraocular metastasis in patients with breast cancer [50,51]. Further research is needed in this field.

This study had some limitations. First, studies used several designs to define associations. Secondly, adjusted covariates differed between the included studies and this might have increased the risk of a confounding bias. Third, cutoff points for the first and last categories varied, and this might have led to study variation.

In conclusion, the present meta-analysis found a significant inverse association between HDL-C and the risk of breast cancer development. TC, TG, LDL-C and ApoA were not found to be significantly correlated with breast cancer development. Hormone use and physical activity affected the relationship of TC with this malignancy, as did race and baseline lipid values for TG. Data on the role of lipids on breast cancer risk are generally conflicting, with low HDL-C usually being related to an increased risk. Further basic and clinical research is required to elucidate the associations between specific cholesterol components and breast cancer risk in certain populations as well as the exact mechanisms underlying the lipid-related signaling pathway involved in the development of this malignancy. Such data will provide evidence on the potential clinical use of lipid-modifying drugs in relation to breast cancer prevention.

## Figures and Tables

**Figure 1 jcm-11-04503-f001:**
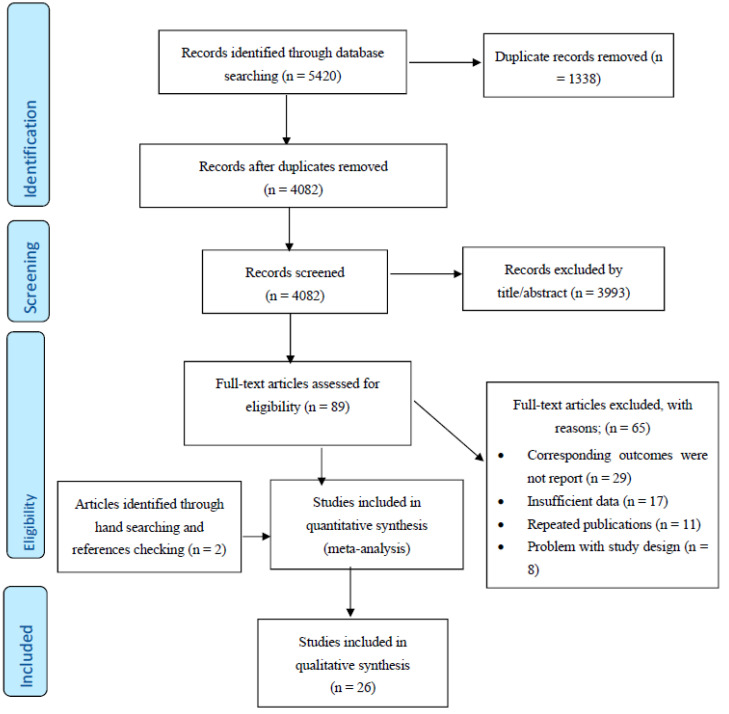
PRISMA (The Preferred Reporting Items for Systematic Reviews and Meta-Analyses) Flow Diagram.

**Figure 2 jcm-11-04503-f002:**
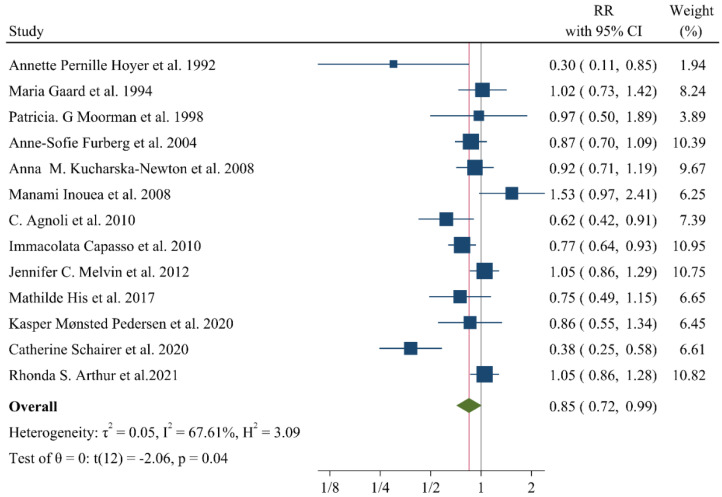
Forest plot of the highest vs. lowest categories of serum HDL–C levels and breast cancer risk. HDL–C: high-density lipoprotein cholesterol [8,9,15,16,20,22,28,32,33,34,35,36,37].

**Figure 3 jcm-11-04503-f003:**
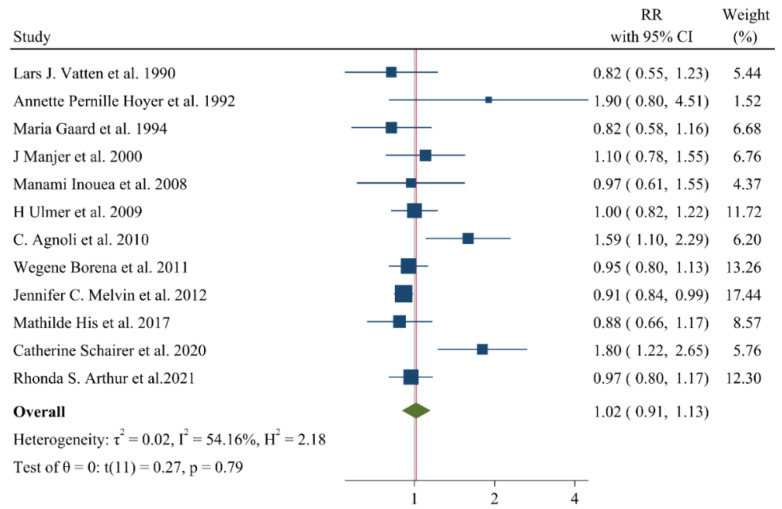
Forest plot of the highest vs. lowest categories of serum triglyceride levels and breast cancer risk [14,15,16,18,22,24,26,28,32,34,36,37].

**Figure 4 jcm-11-04503-f004:**
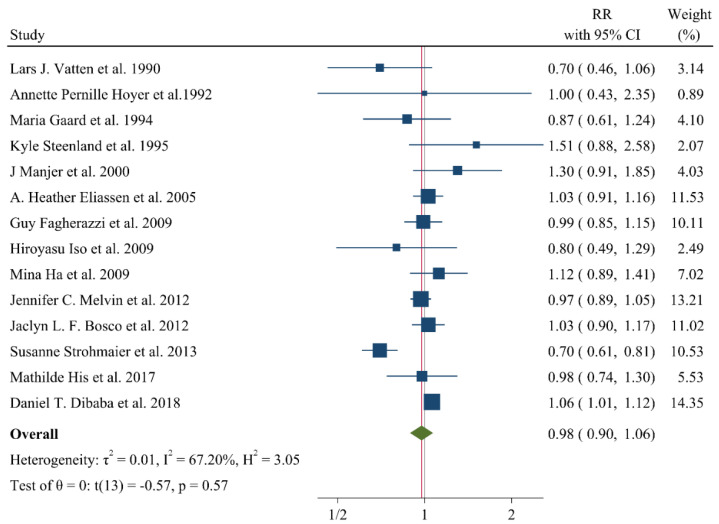
Forest plot of the highest vs. lowest categories of serum total cholesterol levels and breast cancer risk [14,15,16,17,18,19,21,23,25,27,28,29,31,36].

**Figure 5 jcm-11-04503-f005:**
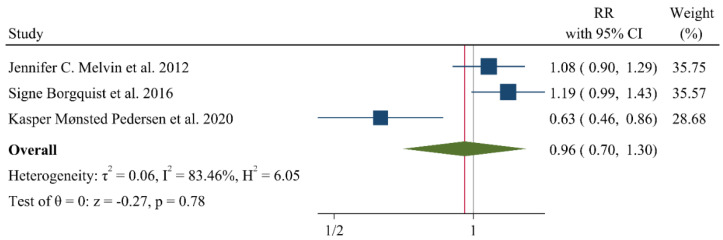
Forest plot of the highest vs. lowest categories of serum Apolipoprotein A levels and breast cancer risk. Apoa: apolipoprotein A [8,28,30].

**Figure 6 jcm-11-04503-f006:**
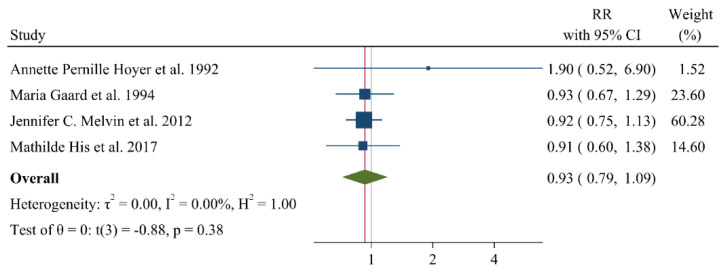
Forest plot of the highest vs. lowest categories of serum LDL–C levels and breast cancer risk. LDL–C: low-density lipoprotein cholesterol [15,16,28,36].

**Figure 7 jcm-11-04503-f007:**
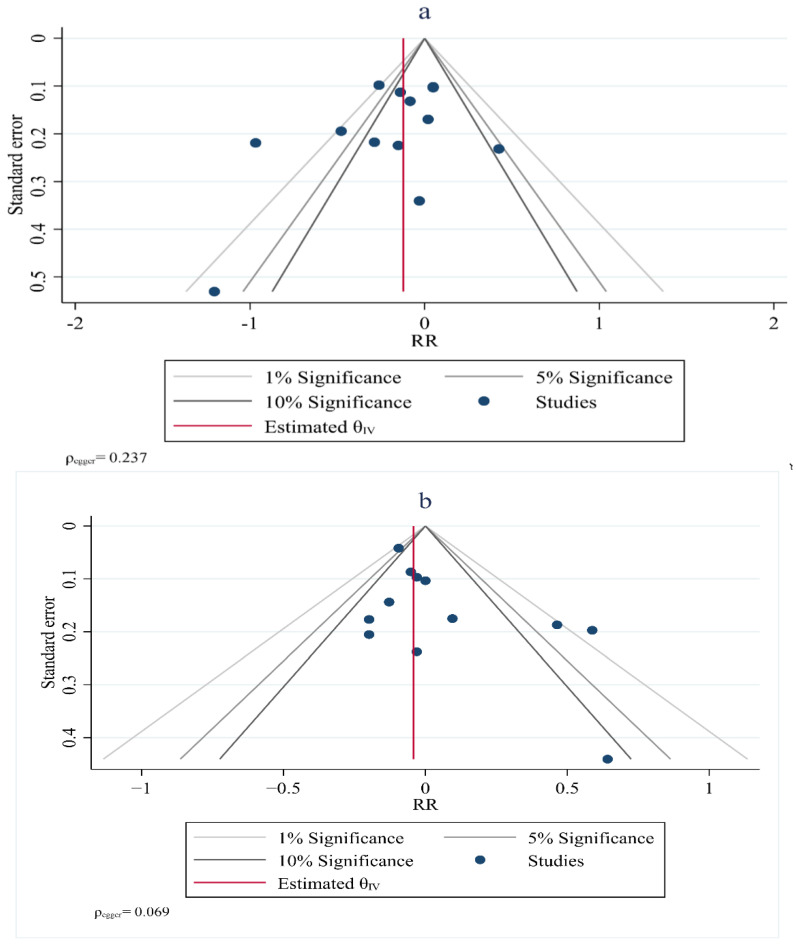
Funnel plot showing lipid profiles (high–density lipoprotein (**a**), triglycerides (**b**), total cholesterol (**c**)) and risk of breast cancer.

## Data Availability

Not applicable.

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
