# Peer review of "Effect of Serum Lipid Profile on the Risk of Breast Cancer: Systematic Review and Meta-Analysis of 1,628,871 Women"

_jcm, 2022, doi:10.3390/jcm11154503_

Round 1
Reviewer 1 Report
Figure 1: The # of screened records are higher than records after duplicate removal. Does that mean 4 of the data sources is repetitive?
Table 1: There are many recent studies show APO was associated with breast cancer risk. However, what was mentioned in the conclusion section is opposite with no further explanation.
In general, the whole idea of sorting out the previous publications is valuable while considering all the different aspects like age, BMI, life style (alcohol intake,smoke,physical activity,and etc).
Author Response
Thanks for your attention in reviewing our manuscript entitled “Effect of Serum Lipid Profile on the Risk of Breast Cancer: Systematic Review and Meta-Analysis of 1,628,871 Women” with the manuscript ID JCM-17295875 that was submitted to the Journal of Clinical Medicine. The revised parts are highlighted in the text and the answers to your valuable comments are as follows:
Figure 1: The # of screened records are higher than records after duplicate removal. Does that mean 4 of the data sources is repetitive?
Reply: we check and then correct it (Page 12).
Table 1: There are many recent studies show APO was associated with breast cancer risk. However, what was mentioned in the conclusion section is opposite with no further explanation.
Reply: according to Table 1, Apo result was significant in Signe Borgquist et al. and Kasper Mønsted Pedersen, but it was not significant in Jennifer C. Melvin et al..
In general, the whole idea of sorting out the previous publications is valuable while considering all the different aspects like age, BMI, life style (alcohol intake, smoke, physical activity, and etc.).

Reviewer 2 Report
The authors present the results of a preregistered systematic review and meta-analysis on the effect of serum lipid profile on breast cancer incidence.
Overall, the manuscript is well written and presents interesting results.
There are some issues that needs to be clarified before acceptance.
- From the paper I understand that a binary dichotomized risk factor (low vs high lipoprotein abundance) was associated with breast cancer risk. Your exlusion criteria also includes "Publications lacking primary data and/or explicit method descriptions, were also excluded".
Does this mean you had patient-level data, or did you use the (potentially different) dichotomizations from the original publications? If you had patient-level data for the N=26 analyzed data sets, how exactly did you select the cut-off value for the dichotomized risk groups (high vs low)?
- Please add the results of the mentioned subgroup analyses (line 152) as a supplement to the manuscript.
- Figure 1 needs to be improved:
- spelling error in word "Corresponnding"
- arrangement of boxes and arrows could be aligned to the phases
- flow categories (left boxes, horizontal orientation) are defined different in PRISMA (Identification, Screening, Included), please adjust accordingly
- Can you give reasons for the exclusions before full-text retrival? (Box: "Records excluded (n = 3993)")
- Figures 2,3,4,5,6 (Forest plots): please annotate exp(ES) either as RR or OR (depending on your selected method in STATA). Also please annotate the direction of the effect as annotation to the x-axis of the forest plot (increased breast cancer risk / decreased breast cancer risk). I assume the reference category for all analyses was low lipoprotein abundance? A vertical reference line at RR=1 (or OR=1, depending on your STATA program) would be very helpful for orientation.
- Please clarify if the NOS score was also used as a selection criterion, or if provided as bias-assessment score without effect on selection.
- How did you handle studies where no breast cancer incidence (missing values in Table 1) were extracted? Would that not be a reason for exclusion? (Patricia. G Moorman et al. USA (1998); Immacolata Capasso et al. Italy (2010) (40); C. Agnoli et al. Italy (2010) (39); Mathilde His et al. France (2017) (41); Catherine Schairer et al. USA (2020) (42))
- Author contributions, Institutional Review Board Statement and supplementary material information, data availability statement, and informed consent paragraphs seems to be from another manuscript or just left with the placeholder.
Author Response
Dear Editor
Thanks for your attention in reviewing our manuscript entitled “Effect of Serum Lipid Profile on the Risk of Breast Cancer: Systematic Review and Meta-Analysis of 1,628,871 Women” with the manuscript ID JCM-17295875 that was submitted to the Journal of Clinical Medicine. The revised parts are highlighted in the text and the answers to your valuable comments are as follows:
Comments and Suggestions for Authors
The authors present the results of a preregistered systematic review and meta-analysis on the effect of serum lipid profile on breast cancer incidence.
Overall, the manuscript is well written and presents interesting results.
There are some issues that needs to be clarified before acceptance.
- From the paper I understand that a binary dichotomized risk factor (low vs high lipoprotein abundance) was associated with breast cancer risk. Your exlusion criteria also includes "Publications lacking primary data and/or explicit method descriptions, were also excluded".
Does this mean you had patient-level data, or did you use the (potentially different) dichotomizations from the original publications? If you had patient-level data for the N=26 analyzed data sets, how exactly did you select the cut-off value for the dichotomized risk groups (high vs low)?
Reply: we used original publications for this meta-analysis.
- Please add the results of the mentioned subgroup analyses (line 152) as a supplement to the manuscript.
Reply: subgroup analysis added to supplementary file.
- Figure 1 needs to be improved:
-Spelling error in word "Corresponnding".
Reply: corrected (Page 12).
-Arrangement of boxes and arrows could be aligned to the phases.
Reply: corrected (Page 12).
-Flow categories (left boxes, horizontal orientation) are defined different in PRISMA (Identification, Screening and Included), please adjust accordingly.
Reply: corrected (Page 12).
- Can you give reasons for the exclusions before full-text retrival? (Box: "Records excluded (n = 3993)").
Reply: corrected (Page 12).
- Figures 2,3,4,5 and 6 (Forest plots): please annotate exp (ES) either as RR or OR (depending on your selected method in STATA). Also please annotate the direction of the effect as annotation to the x-axis of the forest plot (increased breast cancer risk / decreased breast cancer risk). I assume the reference category for all analyses was low lipoprotein abundance? A vertical reference line at RR=1 (or OR=1, depending on your STATA program) would be very helpful for orientation.
Reply: corrected all plots (Page 16-20).
- Please clarify if the NOS score was also used as a selection criterion, or if provided as bias-assessment score without effect on selection.
Reply: NOS score was used as bias-assessment score without effect on selection (Page 4-5, Line 125-126).
- How did you handle studies where no breast cancer incidence (missing values in Table 1) were extracted? Would that not be a reason for exclusion? (Patricia. G Moorman et al. USA (1998); Immacolata Capasso et al. Italy (2010) (40); C. Agnoli et al. Italy (2010) (39); Mathilde His et al. France (2017) (41); Catherine Schairer et al. USA (2020) (42)).
Reply: we can see the referee’s point. We would like to mention that all these 5 studies were nested case-control studies (Page 13-15).
- Author contributions, Institutional Review Board Statement and supplementary material information, data availability statement, and informed consent paragraphs seems to be from another manuscript or just left with the placeholder.
Reply: corrected (Page 8).

Round 2
Reviewer 1 Report
Figure 1's eligibility section makes no sense. The sum of the number of excluded and included studies is incorrect.
Second, it is essential to take into account more factors in this study.
Author Response
Figure 1's eligibility section makes no sense. The sum of the number of excluded and included studies is incorrect.
Replay: we checked it again and correct this part (Page 13).
Second, it is essential to take into account more factors in this study.
Replay: the aim of the present systematic review and meta-analysis was to further investigate the relationship between serum lipid profile and breast cancer development and we assess all the lipid profile variables could added for statistical analysis.

Reviewer 2 Report
Thank you very much for answering my comments and correcting accordingly.
With respect to the methodological question of using highest-vs. lowest (HLM) effect sizes, there may be some more explanation needed to be more specific in the methods part of the paper. The HLM method may have some limitations which were not dicussed in the paper [I cite these here from Bae 2016 (see below), since it is a nice summary already]:
In the HLM method ...
- "information on the quantiles between the lowest and highest ones are ignored"
- "no clear distinction is made between non-intake and low-intake cases in the lowest-intake quantile"
-"no clear cutoff intake level is set for the highest intake quantile."
Please add to your method section if you took any measures to take care of these, and please also discuss the limitations of the HLM method that apply to your study.
Please also add a limitations paragraph to your discussion section to briefly identify the potential impact of your study design, the quality of included studies and decision on methodology, on the results. Such a discussion is missing currently.
A proper understanding of this will be cruicial for the reader to understand the results much better.
--Reference--
Bae JM. Comparison of methods of extracting information for meta-analysis of observational studies in nutritional epidemiology. Epidemiol Health. 2016;38:e2016003. Published 2016 Jan 11. doi:10.4178/epih/e2016003
Author Response
Thank you very much for answering my comments and correcting accordingly.
With respect to the methodological question of using highest-vs. lowest (HLM) effect sizes, there may be some more explanation needed to be more specific in the methods part of the paper. The HLM method may have some limitations which were not discussed in the paper [I cite these here from Bae 2016 (see below), since it is a nice summary already]:
In the HLM method:
- "information on the quantiles between the lowest and highest ones are ignored"
- "no clear distinction is made between non-intake and low-intake cases in the lowest-intake quantile"
-"no clear cutoff intake level is set for the highest intake quantile."
Please add to your method section if you took any measures to take care of these, and please also discuss the limitations of the HLM method that apply to your study.
Replay: we added in statistical part (Page 5, Lines 140-141).
Please also add a limitations paragraph to your discussion section to briefly identify the potential impact of your study design, the quality of included studies and decision on methodology, on the results. Such a discussion is missing currently.
A proper understanding of this will be crucial for the reader to understand the results much better.
Reply: we added a limitation paragraph in the last section of discussion and explain some limitation (Page 8, Line 226-229).
--Reference--
Bae JM. Comparison of methods of extracting information for meta-analysis of observational studies in nutritional epidemiology. Epidemiol Health. 2016;38:e2016003. Published 2016 Jan 11. doi:10.4178/epih/e2016003
